# Clinical and Epidemiological Characteristics of Persistent Bacteremia: A Decadal Observational Study

**DOI:** 10.3390/pathogens12020212

**Published:** 2023-01-29

**Authors:** Shiori Kitaya, Hajime Kanamori, Hiroaki Baba, Kengo Oshima, Kentarou Takei, Issei Seike, Makoto Katsumi, Yukio Katori, Koichi Tokuda

**Affiliations:** 1Department of Infectious Diseases, Internal Medicine, Tohoku University Graduate School of Medicine, Sendai 980-8574, Japan; 2Department of Otolaryngology, Head and Neck Surgery, Tohoku University Graduate School of Medicine, Sendai 980-8574, Japan; 3Department of Intelligent Network for Infection Control, Tohoku University Graduate School of Medicine, Sendai 980-8574, Japan; 4Department of Laboratory Medicine, Tohoku University Hospital, Sendai 980-8574, Japan

**Keywords:** persistent bacteremia, bloodstream infections, follow-up blood cultures

## Abstract

**Background**: Bloodstream infections (BSIs), including persistent bacteremia (PB), are a leading source of morbidity and mortality globally. PB has a higher mortality rate than non- PB, but the clinical aspects of PB in terms of the causative pathogens and the presence of clearance of PB are not well elucidated. Therefore, this study aimed to describe the clinical and epidemiological characteristics of PB in a real-world clinical setting. **Methods**: We performed a retrospective observational survey of patients who underwent blood culture between January 2012 and December 2021 at Tohoku University Hospital. Cases of PB were divided into three groups depending on the causative pathogen: gram-positive cocci (GPC), gram-negative rods (GNRs), and *Candida* spp. For each group, we examined the clinical and epidemiological characteristics of PB, including differences in clinical features depending on the clearance of PB. The main outcome variable was mortality, assessed as early (30-day), late (30–90 day), and 90-day mortality. **Results**: Overall, we identified 31,591 cases of single bacteremia; in 6709 (21.2%) cases, the first blood culture was positive, and in 3124 (46.6%) cases, a follow-up blood culture (FUBC) was performed. Of the cases with FUBCs, 414 (13.2%) were confirmed to be PB. The proportion of PB cases caused by *Candida* spp. was significantly higher (29.6%, 67/226 episodes) than that for GPC (11.1%, 220/1974 episodes, *p* < 0.001) and GNRs (12.1%, 100/824 episodes, *p* < 0.001). The *Candida* spp. group also had the highest late (30–90 day) and 90-day mortality rates. In all three pathogen groups, the subgroup without the clearance of PB tended to have a higher mortality rate than the subgroup with clearance. **Conclusions**: Patients with PB due to *Candida* spp. have a higher late (30–90 day) and 90-day mortality rate than patients with PB due to GPC or GNRs. In patients with PB, FUBCs and confirming the clearance of PB are useful to improve the survival rate.

## 1. Introduction

Bloodstream infections (BSIs) are a leading cause of illness and mortality globally [1]. In the management of patients with BSI, follow-up blood culture (FUBC) is strongly recommended for BSI caused by certain pathogens, such as *Staphylococcus aureus* (*S. aureus*) and *Candida* spp., to confirm the clearance of the BSI and determine the required duration of antimicrobial therapy [2,3]. Conversely, the usefulness of FUBC in patients with BSIs caused by gram-negative rods (GNRs) is controversial [4], and in case of GNRs, clinical judgement is used on an individual basis to assess the necessity of FUBC for bacteremia [5].

The details of persistent bacteremia (PB) have been reported for various causative pathogens, including *S. aureus* [6], GNRs [7], and *Candida* spp. [8]. *S. aureus* is by far the most common cause of PB [9], and PB occurs in 8% to 39% of cases of *S. aureus* bacteremia [10]. *S. aureus* PB suggests the presence of specific microbiological features (e.g., antibiotic tolerance), difficult-to-eradicate sites of infection (e.g., abscesses), or patient-related factors that impede the rapid clearance of bacteria from the blood [10]. Furthermore, *S. aureus* PB is associated with an increased risk of embolic complications, prolonged length of hospital stays, and higher mortality [11]. As regards GNR-related infections, uncomplicated GNR bacteremia is generally transient, but it can become persistent if the source is not controlled or metastatic infection occurs [12]. One study found that more than one third of patients with GNR bacteremia in the intensive care units (ICUs) develop GNR PB because they have more severe clinical conditions that frequently require the placement of indwelling endovascular devices and the invasive support of vital functions [13]. PB due to *Candida* spp. has been reported in 8% to 15% of patients with candidemia [14]. In addition to host factors such as poor general condition, several possible mechanisms have been suggested for *Candida* spp. PB, such as: drug resistance, low serum drug levels, endovascular infection, deep-tissue abscesses, and infections associated with prosthetics [14].

Although much research has examined PB, many studies have focused on a single bacterial species, e.g., *S. aureus* [6], *Enterococcus faecium* [15], *Pseudomonas aeruginosa* [16], and *Klebsiella pneumoniae* [17]. A number of studies have evaluated groups of pathogens, such as GNRs [7] and *Candida* spp. [8], but few have investigated the differences in clinical characteristics between multiple bacterial categories. In addition, studies have largely focused on risk factors and not so much on outcomes, and no study has compared clinical characteristics of patients with and without the clearance of PB. 

A finding confirming that clinical outcomes are related to the pathogen causing PB and the clearance of PB would lead to the appropriate management of the condition and may help to improve the survival rate of patients. Therefore, we investigated differences in clinical characteristics of PB between three groups of bacterial species (gram-positive cocci [GPC], GNRs, and *Candida* spp.) and differences in mortality depending on the clearance of PB.

## 2. Methods

### 2.1. Study Design and Setting

This was a retrospective, single-center, observational study performed at Tohoku University Hospital, Sendai, Miyagi, Japan. We collected study variables by reviewing electronic clinical charts and the hospital records of patients who underwent blood culture (BC) at the hospital between January 2012 and December 2021. Almost all BC data collected at our hospital are sent for diagnosis to the Department of Infectious Diseases, which has advised the other hospital departments on treatment methods and the duration of antibiotic therapy since 2012. In PB, the department recommends appropriate source control, such as incision drainage or the removal of catheters, when the focus of the infection is known and the identification of the focus by imaging methods, such as computerized tomography scan or gallium scintigraphy, when it is not.

For this study, the data of each bloodstream isolate were collected from the computerized records of the Department of Laboratory Medicine, and the following anamnestic and clinical data were obtained from the medical records and the database of the Department of Infectious Diseases: sex, age, comorbidities, body mass index, and body temperature; blood test results, including the levels of serum white blood cells, neutrophils, and C-reactive protein; the presence of intravascular devices, i.e., a central line such as a conventional central venous catheter (CVC), peripheral inserted central catheter, tunneled central venous catheter, and implanted central venous port (CV port), and the removal of such devices; the presence of cardiac implantable electronic devices (CIEDs), such as cardiac pacemakers, implantable cardioverter defibrillators, and devices for cardiac resynchronization therapy; the presence of a ventricular assist device; a history of valve replacement; the presence of a vascular graft; the use of extracorporeal membrane oxygenation (ECMO), continuous hemodiafiltration, and mechanical ventilation; the interval between initial BC and FUBC; the duration of bacteremia; site of infections; duration of hospitalization; whether time was spent in an ICU, high care unit, or coronary care unit between the initial BC and the last FUBC; antibiotic use; the source control performance; and mortality, recorded as early (30-day), late (30–90 day), and 90-day mortality. 

All patients aged 18 years and older were eligible for inclusion if they were diagnosed with PB. BSI was defined as one or more positive BCs, and PB was defined as the identification of the same microorganism species in the BCs of the same individual on two or more consecutive instances. The exclusion criteria included polymicrobial PB and possible contaminants (such as coagulase-negative staphylococci [CoNS], *Propionibacterium* spp., and *Corynebacterium* spp.). Clinical characteristics were examined retrospectively using electronic clinical charts, hospital records, and microbiological data. 

The Human Ethical and Clinical Trial Committee of Tohoku University Hospital approved this study (2019-1-270). Patient consent requirements were waived because of the retrospective nature of the study.

### 2.2. Definitions and Outcomes 

FUBC was defined as BC performed within 7 days of the initial positive BC. If the same species of microorganism as the original one was found in at least one pair of FUBCs, the case was classified as FUBC positive [18]. However, if different species were identified in the FUBC, the bacteremia was considered to be a new case and was excluded from the analysis [18]. The source of bacteremia was determined by reviewing the clinical chart from the Department of Infectious Diseases specialists. Infectious disease physicians treating patients with bacteremia identify the foci of infections by considering the species detected by the BC; medical records; clinical findings; imaging studies; and microbiological data. The site of infection was defined by applying the Centers for Disease Control and Prevention criteria [19]. Primary bacteremia was defined as a positive BC in a patient in whom no focus of infection had been identified after an extensive clinical, radiological, and microbiological evaluation [13]. If a BC was positive for *Corynebacterium* spp., *Bacillus* spp., or *Propionibacterium acnes*, the contamination level was assessed; contamination was defined as the isolation of CoNS from one of the two sets of BCs [20]. In instances where the same pathogenic agent was cultured across multiple BCs, a diagnosis of bacteremia was confirmed. If a pathogen was identified in only one of multiple sets of BCs, bacteremia was diagnosed only in the case of non-contamination. If more than one microorganism was identified from a single BC, all microorganisms were considered separately. If multiple BCs were performed for a patient and a negative BC occurred between two positive ones, the patient was considered to have separate cases of bacteremia. For PB, multiple consecutive BCs that identified the same strain were counted as one episode. The single counts represent the number of BCs itself, and the episodes represent a series of BSI episodes. The duration of bacteremia was defined as the number of days between the first positive BC and the last positive FUBC [6]. The clearance of PB was defined as a negative result from the final BC. Source control was defined as any procedure performed to control the source of the bacteremia after the initial BC, e.g., the removal of intravascular or endovascular devices, the surgical debridement of the skin, the drainage or removal of an intra-abdominal abscess, and the drainage of a dilated bile duct or hydronephrosis [21]. Renal failure was defined as a case glomerular filtration rate of less than 60 mL/min per 1.73 m^2^ [22]. Neutropenia occurring during a BSI was defined as an absolute neutrophil count of less than 500/μL according to the common terminology criteria for adverse events version 5.0 [23]. Immunosuppression was considered to be present in patients with neutropenia, hematopoietic stem-cell transplantation, solid organ transplantation, or corticosteroid therapy (prednisone 16 mg/day for 15 days) [18]. Respiratory failure was defined as patients with SpO_2_ less than 90% or oxygen administration [24]. Shock was defined as a patient with a systolic blood pressure less than 90 mmHg or requiring vasopressor support [25].

Antimicrobial therapy was considered to be inappropriate when at least one of the following conditions was met: the administration of ineffective antimicrobial agents, i.e., agents that did not effectively treat the organisms identified in the BC; the continuation of the initial antimicrobial agents even though the result of the susceptibility test was known and de-escalation was possible [26]; and the administration of antibiotic therapy for a shorter time than specified by current medical standards [27]. In all other cases, antimicrobial therapy was considered to be appropriate. 

The primary outcome variable was early (30-day), late (30–90 day) and 90-day mortality (day 0 was the day of the initial BC). 

### 2.3. Methods for Collecting Blood Samples for Culture

During the study period, arterial or venous blood was aseptically obtained from patients with suspected BSI and inoculated into aerobic and anaerobic BACT/ALERT^®^ FA plus bottles (BioMérieux, Durham, NC, USA). Each bottle was incubated in a BACT/ALERT VIRTUO instrument (BioMérieux, Durham, North Carolina) at 37.0 °C for seven days. A VITEK MS system (BioMérieux, Métropole de Lyon, France) and a Walk Away 96 Plus system (Siemens Healthcare Diagnostics, Deerfield, IL, USA) were used for identification and susceptibility testing. For fungi and *Haemophilus influenzae*, the susceptibility tests were performed with a RAISUS S4 device (Nissui Pharma, Tokyo, Japan).

### 2.4. Statistical Analysis

We compared differences in clinical characteristics according to the microbial pathogen, including GPC, GNRs, and *Candida* spp., and differences in mortality rates in terms of the success or failure with regards to clearing PB due to each pathogen. 

To compare differences in the characteristics of PB due to GPC, GNRs, and *Candida* spp., we analyzed continuous variables with a Kruskal-Wallis test followed by a Steel-Dwass test and categorical variables with Fisher’s exact test. *p*-values were corrected with the Hyan-Holm step-down Bonferroni procedure. To compare the two subgroups with and without the clearance of PB for each pathogen, we compared the means of continuous variables with the Mann-Whitney U test and proportions of categorical variables with Fisher’s exact test. Continuous variables are presented as medians (interquartile range, IQR). The analysis was performed with JMP Pro 16 statistical analysis software (SAS Institute, Cary, NC, USA). Differences were considered significant at a corrected *p*-value of less than 0.050.

## 3. Results and Discussion

### 3.1. Patient Selection and Causative Pathogens of PB

The details regarding patient selection are shown in Figure 1. A total of 31,591 BCs were performed at Tohoku University Hospital from January 2012 to October 2021 and were screened for inclusion. Among the 6709 cases with a positive result at the initial BC, 3124 (46.6%) underwent FUBCs. Of the FUBCs performed, 414 (13.2%) were PB. The proportion of cases of PB due to *Candida* spp. was significantly higher (29.6%, 67/226 episodes) than the proportion due to GPC (11.1%, 220/1974 episodes, *p* < 0.001) and GNRs (12.1%, 100/824 episodes, *p* < 0.001). People aged 60 years and older accounted for 56.0% of all cases, and men accounted for 62.1%. The causative pathogens of PB are shown in Table 1.

### 3.2. Differences in the Clinical Characteristics of PB Caused by GPC, GNRs, and Candida spp. 

We found no significant differences between the three pathogen groups in terms of vital signs, laboratory markers, the use of source control, or the appropriateness of antibiotic use. The clinical characteristics of PB in the three groups are described in detail below and shown in Table 2.

#### 3.2.1. Clinical Characteristics of PB Caused by GPC

In terms of comorbidities, the proportion of patients with end-stage renal disease receiving hemodialysis tended to be higher in the GPC group (24 cases, 10.9%) than in the GNR (6 cases, 6.0%, *p* = 0.430) and *Candida* spp. groups (1 case, 1.5%, *p* = 0.039). PB is common in dialysis patients because they are prone to vascular access-associated BSI; this infection most frequently occurs in patients who use dialysis catheters, and the catheter is reported to be the source of BSI in 48% to 73% of all cases of BSI in hemodialysis patients [28]. GPC are well known as the causative pathogens of hemodialysis catheter-related bloodstream infection (CRBSI) [29], and hemodialysis has been individually associated with recurrent BSI caused by *S. aureus* [30]. This study included many dialysis patients with GPC-induced PB. 

The CIED implantation rate was higher in the GPC group (22 cases, 10.0%) than in the GNR (five cases, 5.0%, *p* = 0.192) and *Candida* spp. groups (0 case, 0%, *p* = 0.010). CIEDs are increasingly being used in cardiac disease management, but CIED infection is a serious complication and associated with substantial morbidity, mortality, and healthcare costs [31]. Like other implantable devices, CIEDs are said to have a high risk of infection; infection was found to occur in approximately 1% to 1.3% of cases [31]. In an earlier cohort study, CIED infections were characterized by a predominance of GPC, with CoNS (42–69%) representing the most commonly isolated bacteria, followed by *S. aureus* (14–29%) [32]. In cases where a long time has passed since CIED implantation, the lead may adhere to the blood vessel, which may make it difficult to extract. Furthermore, removal can be too invasive, especially in older patients or those with significant comorbidities [33]. The high number of cases with difficulty in source control may be linked to the high number of GPC-induced PB in CIED infections. 

Infectious endocarditis (IE) was significantly more common in the GPC group (31 cases, 11.1%) than in the GNR (2 cases, 1.7%, *p* = 0.004) and *Candida* spp. groups (one case, 1.3%, *p* = 0.007). The epidemiology of IE has gradually changed over the years, and healthcare-associated IE now accounts for 25% to 30% of cases because the use of intravenous lines and intracardiac devices has increased [34]. IE associated with these indwelling devices may be difficult to treat with antimicrobials alone. Furthermore, when bacteria attached to warts or indwelling devices are distributed homogenously throughout the body, they can cause several complications, such as abscesses, pyogenic spondylitis, and septic embolization [35]. Earlier studies showed that the emergence of healthcare-associated IE is characterized by a predominance of GPC, i.e., *S. aureus*, CoNS, and *Enterococci* spp. [36]. In this investigation, methicillin-susceptible *S. aureus* (MSSA) (nine cases, 26.4%), *Enterococcus faecalis* (six cases, 17.6%), and methicillin-resistant *S. aureus* (MRSA) (five cases, 14.7%) were the most prevalent causative agents of IE, in concordance with previously published research. Thus, in the present study, IE is thought to have been a common cause of GPC PB because bacteria were present throughout the body and easily formed lesions. 

Pyogenic spondylitis tended to be more common in the GPC group (14 cases, 5.0%) than in the GNR (one case, 0.9%, *p* = 0.155) and *Candida* spp. groups (0 case, 0%, *p* = 0.138). Pyogenic spondylitis involves the intervertebral disc and adjacent vertebrae and mainly occurs by the hematogenous spread of bacteria from a distant site [37]. Therefore, pyogenic spondylitis is often complicated by BSI, and it is highly associated with type 2 diabetes, IE, chronic kidney disease, cancer, immunosuppressive disorders, and intravenous drug abuse [38,39]. Different microbial etiologies of pyogenic spondylitis have been reported in different studies, but the most commonly reported etiologic species is GPC, i.e., *S. aureus* followed by *Streptococcus* spp. [40], a finding that agrees with the results of the present study.

#### 3.2.2. Clinical Characteristics of PB Caused by GNRs

Insertion rates of vascular grafts tended to be higher in the GNR group (20 cases, 20.0%) than in the GPC (16 cases, 7.3%, *p* = 0.005) and *Candida* spp. groups (6 cases, 9.0%, *p* = 0.160). Infection involving prosthetic grafts is one of the most serious complications after aortic surgery and is generally associated with a poor prognosis. The incidence of infection in abdominal and thoracic aortic endografts ranges from 0.2% to 5%, and an infected endograft can manifest as BSI, graft disruption, or hemorrhage into adjacent organs [41,42,43]. The identification of an infected endograft is essential when there is active bleeding or a risk of BSI, but surgical procedures are often difficult, depending on the patient’s general condition, life expectancy, and operative risk [44]. Therefore, patients with vascular grafts are deemed be at high risk of vascular graft infection and PB. Previous studies found that the common organisms of vascular graft infection are *S. aureus*, *Staphylococcus epidermidis*, and Gram-negative enteric organisms, such as *Escherichia coli*, *Pseudomonas* spp., *Proteus* spp., and *Klebsiella* spp. [45,46]. Our study included patients who did not have a vascular graft infection despite having a vascular graft, but the frequency of each strain in those with PB was similar to that found in previous studies.

The proportion of patients with respiratory failure and the use of mechanical ventilation were higher in the GNR group (44 cases, 44.4%; and 43 cases, 43.0%, respectively) than in the GPC (61 cases, 27.7%, *p* = 0.014; and 57 cases, 25.9%, *p* = 0.008, respectively) and *Candida* spp. groups (17 cases, 25.4%, *p* = 0.030; and 21 cases, 31.3%, *p* = 0.292, respectively). Patients with respiratory failure are more frequently managed by tracheal intubation, which increases the risk of ventilator-associated pneumonia (VAP) [47]. VAP remains one of the most common infections in patients requiring invasive mechanical ventilation. About 10% of patients put on mechanical ventilation develop VAP [41], and the mortality rate in VAP has been estimated to be 13% [42]. In addition to having decreased respiratory function, patients under mechanical ventilation management often are in poor general condition and have poor circulatory dynamics, for example. Therefore, a CVC is frequently inserted for administration of drugs, e.g., vasopressor agents, and indwelling urethral catheters are also frequently inserted in patients experiencing difficulty urinating. As a result, in addition to VAP, patients under mechanical ventilation management also have a high risk of CRBSI and catheter-associated urinary tract infections, with there being a high possibility that such infections will lead to BSI. Ventilated patients have a higher prevalence of aerobic GNRs, and the most common pathogens of VAP are *Escherichia coli* and *Klebsiella* spp. [48]. A recent multicenter observational study reported that the presence of ventilatory support was not significantly correlated with GNR PB [21]; however, there is no doubt that GNRs can cause VAP and also bacteremia. 

Urinary tract infections (UTIs), surgical site infection (SSIs), and primary bacteremia tended to be more common in the GNR group (10 cases, 8.6%; 7 cases, 6.0%; and 25 cases, 21.6%, respectively) than in the GPC (2 cases, 0.7%, *p* < 0.001; 6 cases, 2.1%, *p* = 0.124; and 27 cases, 9.6%, *p* = 0.008, respectively) and *Candida* spp. groups (2 cases, 2.5%, *p* = 0.255; 0 cases, 0%, *p* = 0.128; and 10 cases, 12.5%, *p* = 0.259, respectively). Our hospital is a tertiary emergency hospital, so we treat many patients who are undergoing urological operations, have indwelling urethral catheters for hydration during chemotherapy, or have difficulty urinating as a result of their poor general condition or the physical obstruction of the urinary tract due to tumors or kidney stones. The long-term placement of urethral catheters is a well-known risk factor for catheter-associated urinary tract infections, which often lead to secondary BSI [43]. In complicated cases of UTI, the common causative agents are GNRs, i.e., *Escherichia coli*, followed by *Enterococci* spp. and *Klebsiella pneumoniae* [49]. Therefore, the finding that the cases with GNR PB in our hospital included many patients with a UTI is consistent with the previously reported research. 

The causative pathogens of SSI after abdominal surgery are often GNRs, such as *Escherichia coli*, *Acinetobacter baumannii*, and *Pseudomonas* spp. [44]. On the other hand, after cardiovascular surgery, in general the most common cause of SSIs (i.e., in approximately 80% of all cases) are GPC, especially *S. aureus* [44]. In our hospital, GNRs (*Enterobacter cloacae*, *Klebsiella oxytoca*, and *Burkholderia cepacia*) were the causative pathogens of SSIs in a large proportion of patients after cardiovascular surgery (three cases, 60.0%). Our hospital performs a large number of cardiovascular operations, and the fact that GNRs are a frequent cause of PB caused by SSIs after cardiovascular surgery is considered to be one of the reasons why GNRs were a frequent cause of PB caused by SSIs in our hospital.

The ICU admission rate tended to be higher in the GNR group (57 cases, 57.0%) than in the GPC (86 cases, 39.1%, *p* = 0.011) and *Candida* spp. groups (28 cases, 41.8%, *p* = 0.119). The ICU stay was also significantly longer in the GNR group (median, 4.0; IQR, 0–45.0) than in the GPC (median, 0; IQR, 0–10.3; *p* < 0.001) and *Candida* spp. groups (median, 0; IQR, 0–15.0; *p* = 0.034). In this study, CRBSI was the most common focus of infection among patients admitted to the ICU (88 cases, 39.6%), followed by primary bacteremia (37 cases, 16.7%), and endovascular device infections (19 cases, 8.6%). When we examined the causative pathogens of PB, we found that patients admitted to the ICU had a higher proportion of GNR infections than those not admitted to the ICU among the cases of CRBSI (21 cases, 23.9%, vs. 15 cases, 12.1%, respectively), primary bacteremia (15 cases, 40.5% vs. nine cases, 27.3%, respectively), and endovascular device infections (5 cases, 26.3% vs. 0 cases, 0%, respectively). GNRs, especially antibiotic-resistant GNRs, have been reported to account for a higher proportion of BSI-causing pathogens in patients admitted to the ICU than in patients not admitted [50]. At our hospital, approximately 70% of patients admitted to the ICU are postoperative patients after cardiovascular surgery. Such patients are often intubated and artificially ventilated, which involves frequent aspiration procedures for the prevention and treatment of VAP [51]. The dispersal of sputum containing GNR through the aspiration procedure may have contributed to the higher rate of healthcare-associated infections caused by GNRs in the ICU. Additionally, it is feasible that GNR-laden sputum dispersed by suction and adhering to surgical incisions may have played a role in the high incidence of GNRs as a causative agent of postoperative cardiovascular infections in our hospital.

#### 3.2.3. Clinical Characteristics of PB Caused by Candida spp.

Intravascular devices were inserted significantly more often in the *Candida* spp. group (56 cases, 83.6%) than in the GPC (137 cases, 62.3%, *p* = 0.003) and GNR groups (64 cases, 64.0%, *p* = 0.016). Regarding the site of infection, CRBSI was significantly more common in the *Candida* spp. group (47 cases, 58.8%) than in the GPC (115 cases, 40.9%, *p* = 0.010) and GNR groups (37 cases, 31.9%, *p* = 0.001). Intravascular devices are often used for the administration of drugs in patients receiving chemotherapy and nutrients in patients experiencing difficulty with oral intake. Patients with intravascular devices are often in poor general condition, e.g., because of cancer, immunosuppression, or postoperative conditions, making them more susceptible to infection than patients without intravascular devices. In immunosuppressed patients, *Candida* spp. are more likely to cause infection, and the prolonged use of intravascular devices is a risk for CRBSI [52]. A retrospective cohort study showed that in recurrent CRBSI, *Candida albicans* was the most common pathogen (44 cases, 29.1%), followed by *S. aureus* (21 cases, 13.9%) [53]. Furthermore, the study also showed that fungal CRBSIs were associated with recurrent catheter-related infections even if the infected catheters were removed and appropriate antibiotic administration was implemented. In the present study, in 42/47 cases (89.4% of cases), CRBSI caused by *Candida* spp. resulted in PB, despite appropriate source control such as the removal or exchange of the CVC/CV port. Our results are consistent with those of the above-mentioned study in that they show that it is difficult to eliminate CRBSI caused by *Candida* spp. even with appropriate source control. 

The use of ECMO tended to be higher in the *Candida* spp. group (seven cases, 10.4%) than in the GPC (three cases, 1.4%, *p* = 0.006) and GNR groups (three cases, 3.0%, *p* = 0.182). Nosocomial infection frequently occurs in patients supported by ECMO, with reported rates ranging from 8% to 64% [54]. The most frequently reported nosocomial infections in such patients are VAP and BSI [55]. Patients supported by ECMO generally have refractory cardiogenic shock or refractory acute respiratory distress syndrome, making early withdrawal from ECMO often difficult. In addition, frequent replacement is often difficult because the ECMO must replace the entire circuit rather than just the catheter. These factors prolong the catheter retention period and often prevent proper source control in case of CRBSI, which is thought to result in many cases of PB. Patients under ECMO management are often admitted to the ICU, and *Candida* spp. were reported to often be the causative pathogen of BSI in patients admitted to the ICU [56]. In the present study, we found that CRBSI and primary bacteremia were the most common infection foci in patients undergoing ECMO (5/13 of all infection focus in patients undergoing ECMO, 38.5% each). Previous studies showed that, after CoNS, *Candida* spp. are the second most common causal infectious agents during ECMO [57] and that the most common organisms colonizing an ECMO catheter in primary BSI were *Candida* spp. and GPC [57]. These results also support our finding that *Candida* PB was more common than other types of PB among patients undergoing ECMO.

The period until FUBC and the duration of bacteremia were significantly longer in the *Candida* spp. group (median, 4.0 days [IQR, 2.0–5.5], and median, 5.0 days [IQR, 3.0–7.5], respectively) than in the GPC group (median, 3.0 days [IQR, 2.0–4.0], *p* = 0.016, and median, 3.0 days [IQR, 2.0-7.0], *p* = 0.002, respectively) and GNR groups (median, 3.0 days [IQR, 1.0–4.0], *p* = 0.028, and median, 4.0 days [IQR, 2.0–7.0], *p* = 0.019, respectively). In general, BCs for yeasts require a long period of incubation (>24 h) [58]; as a consequence, in candidemia, the identification of the pathogens in the first BC was delayed, which may have delayed FUBC. A prolonged incubation time may also cause delays in clinical intervention. Furthermore, although persistent candidemia may indicate treatment failure, *Candida* spp. are known to take a long time to clear from the bloodstream even when the treatment is effective, resulting in a longer duration of PB.

### 3.3. Differences in the Mortality Rates of PB Caused by GPC, GNRs, and Candida spp.

Late (30–90 day) and 90-day mortality rates were significantly higher in the *Candida* spp. group (22 cases, 32.8%, and 23 cases, 34.3%, respectively) than in the GPC (21 cases, 9.5%, *p* < 0.001, and 35 cases, 15.9%, *p* = 0.005, respectively) and GNR groups (10 cases, 10.0%, *p* = 0.001, and 17 cases, 17.0%, *p* = 0.031, respectively) (Figure 2). A previous study provided evidence that *Candida* spp. is the causative pathogen of infection in patients with immunosuppressive conditions associated with chemotherapy or organ transplantation for malignant cancers and showed that the proportion of patients with solid or blood cancers and immunosuppressive status is higher among patients with *Candida* PB than among those with GPC PB and GNR PB [52]. In candidemia, the fungus spreads homogeneously, which can cause conditions such as IE [59], purulent spondylitis, and purulent arthritis [60]. Patients with candidemia are often in poor general condition because of immunosuppression or other reasons, so active source control such as debridement is often difficult. Therefore, candidemia is thought to be more likely to lead to PB, and we suggest that difficulties in source control and longer periods of BSI may have led to the associated increased mortality.

### 3.4. Differences in Clinical Characteristics Depending on the Clearance of PB 

In Figure 3, the mortality of patients with PB due to each pathogen is presented separately for the subgroups with and without the clearance of PB. In all pathogen groups, ie, GPC, GNRs, and *Candida* spp., the subgroup without clearance of PB tended to have a higher mortality rate than the subgroup with clearance. In GNR PB, the subgroup without clearance had significantly higher early (30-day; five cases, 17.9%), late (30–90 day; 7 cases, 25.0%), and 90-day mortality (12 cases, 42.9%) than the subgroup with clearance of PB (2 cases, 2.8%, *p* = 0.018, 3 cases, 4.2%, *p* = 0.005 and 5 cases, 6.9%, *p* < 0.001, respectively). In the group with *Candida* PB, late (30-90 day) and 90-day mortality were significantly higher in the subgroup without clearance of PB (14 cases, 66.7%, and 15 cases, 71.4%, respectively) than in the subgroup with clearance (8 cases, 17.4%, *p* < 0.001, and 8 cases, 17.4%, *p* < 0.001, respectively). In GPC PB, only early (30-day) mortality was significantly higher in the subgroup without clearance of PB (7 cases, 14.6%) than in the subgroup with clearance (16 cases, 9.3%, *p* = 0.015). Previous studies reported that patients with *S. aureus* PB [6] and *Candida* PB [8] had a higher mortality rate than those with non-PB due to *S. aureus* and *Candida*. In addition, for GNRs, the mortality rate in patients with PB is reported to be twice that in patients with non-PB [7]. These findings suggest that PB is a prognostic factor for poor outcome in patients with bacteremia, regardless of the pathogen. Therefore, in the case of PB, performing FUBC to confirm the clearance of PB is essential for improving the survival rate. 

An overview of the differences in the clinical characteristics depending on the clearance of PB is shown in Table 3. If we consider the clinical features other than mortality, the data show no significant difference in the frequency of source control and the proper use of antibiotics between patients with and without the clearance of PB. However, significant differences between the subgroups were found for IE in the GPC PB group and immunosuppression and ECMO in the *Candida* PB group. In our hospital, when a BC indicates bacteremia, the Department of Infectious Diseases intervenes in the diagnosis and advises on antibiotic selection, treatment duration, and source control implementation. This process may explain why we found no difference in the frequency of source control and the use of appropriate antibiotics between the subgroups with and without the clearance of PB. IE, the use of ECMO, and immunosuppression are considered to be major patient-related risk factors for PB. The present study indicates that these patient-related characteristics may affect the clearance of PB even when appropriate treatment is performed.

## 4. Limitations

Our study has two limitations. First, the findings may not apply to all patients with BSI because the study was performed at a single university hospital in Japan. Nevertheless, such long-term, large-scale studies of PB are uncommon. Secondly, the data contained within the medical records was insufficient to fully elucidate the underlying reasons for the decision to either perform or abstain from performing FUBC.

## 5. Conclusions

To our knowledge, this was the first study to investigate the differences in clinical and epidemiological aspects of PB in relation to the pathogen and the presence or absence of clearance of PB for each pathogen group. The key conclusions of the study were as follows; (1) GPC are the most common cause of PB in our hospital, and GPC PB is more frequent in patients on hemodialysis for end-stage renal disease and in those with CIED implantation, IE, and pyogenic spondylitis; (2) GNR PB is more frequent in patients with inserted vascular grafts, respiratory failure, the use of mechanical ventilation, UTI, or SSI, or primary bacteremia and in those admitted to an ICU and is associated with a longer stay in the ICU; (3) *Candida* spp. PB is more frequent in patients with intravascular devices, those on ECMO, and those with CRBSI; (4) patients with *Candida* spp. PB tend to have a higher mortality rate and longer duration of bacteremia than those with GPC PB and GNR PB, and so to increase the survival rate, aggressive source control should be performed in addition to the use of antifungal drugs; and (5) in PB, performing FUBC and confirming the clearance of PB can help to improve the survival rate.

## Figures and Tables

**Figure 1 pathogens-12-00212-f001:**
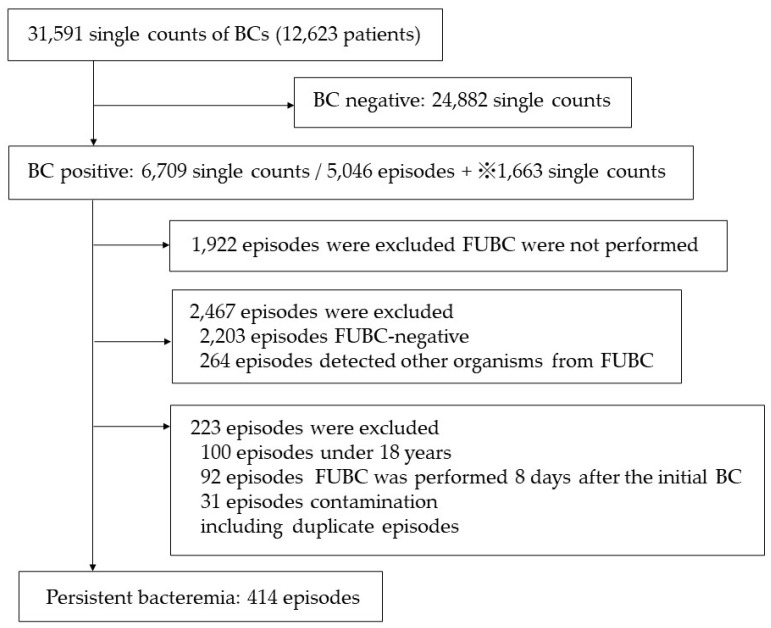
Patient selection. BC, blood culture; FUBC, follow-up BC. If a microorganism of the same species as the initial one was detected in at least one pair of FUBCs, the case was classified as FUBC positive. Conversely, if disparate species were identified in the FUBC, the episode of bacteremia was deemed to be a novel occurrence. The single counts represent the number of BCs itself, and the episodes represent a series of bloodstream infection episodes. ※1663 single counts, including second and subsequent positive BCs in cases of persistent bacteremia and second and subsequent positive BCs in cases in which FUBC detected a different strain than the first BC.

**Figure 2 pathogens-12-00212-f002:**
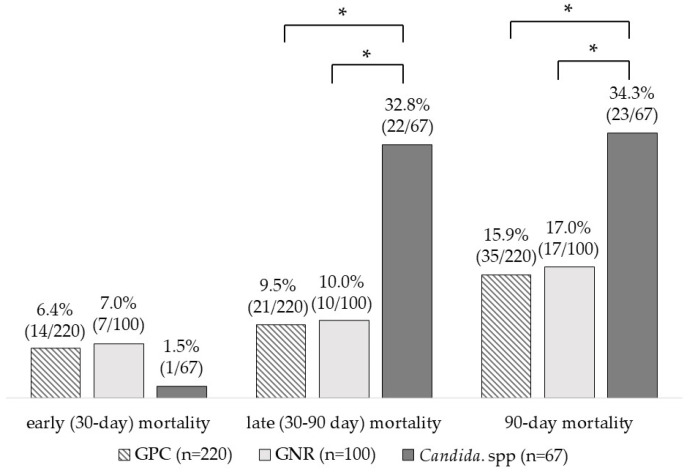
Mortality of patients with persistent bacteremia due to gram-positive cocci, gram-negative rods, and *Candida* species. GPC, gram-positive cocci; GNRs, gram-negative rods. * *p* < 0.050.

**Figure 3 pathogens-12-00212-f003:**
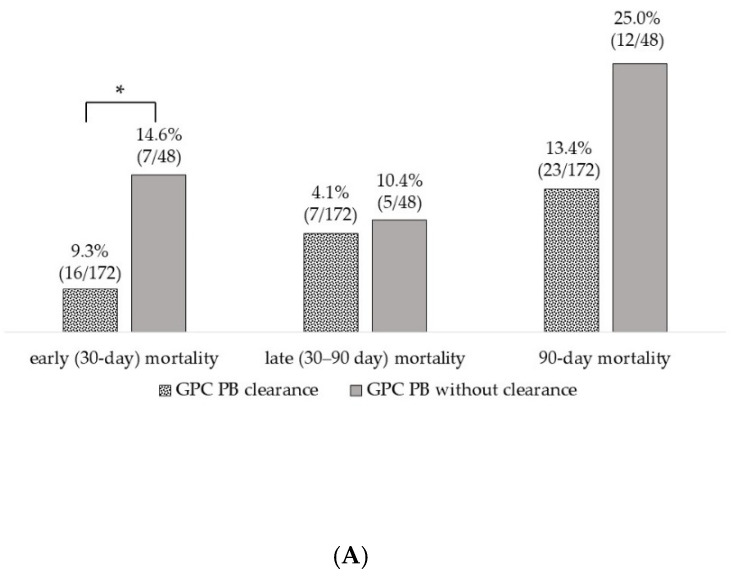
Differences in mortality between patients with and without clearance of persistent bacteremia. (**A**) persistent bacteremia (PB) due to gram-positive cocci; (**B**) PB due to gram-negative rods; (**C**) PB due to *Candida* spp. GPC, gram-positive cocci; GNR, gram-negative rods; PB, persistent bacteremia. * *p* < 0.050, ** *p* < 0.010.

**Table 1 pathogens-12-00212-t001:** Incidence of causative pathogens.

Causative Organisms	Number	Frequency (%)
**Gram-positive coccus**		
*Staphylococcus epidermidis*	62	15.0
*Staphylococcus aureus* (MSSA)	49	11.8
*Staphylococcus aureus* (MRSA)	43	10.4
*Enterococcus faecalis*	17	4.1
*Staphylococcus hominis*	8	1.9
*Staphylococcus lugdunensis*	6	1.4
*Enterococcus faecium*	6	1.4
*Staphylococcus capitis*	4	1.0
*Streptococcus gordonii*	3	0.7
*Streptococcus anginosus*	3	0.7
*Staphylococcus caprae*	3	0.7
*Streptococcus agalactiae*	2	0.5
*Staphylococcus warneri*	2	0.5
*Staphylococcus haemolyticus*	2	0.5
*Staphylococcus caprae*	2	0.5
*Abiotrophia defectiva*	1	0.2
*Granulicatella adiacens*	1	0.2
*Kocuria rhizophila*	1	0.2
*Peptostreptococcus micros*	1	0.2
*Staphylococcus piscifermentans*	1	0.2
*Streptococcus gallolyticus*	1	0.2
*Streptococcus pasteurianus*	1	0.2
*Streptococcus sanguinis*	1	0.2
**Gram-positive bacilli**		
*Bacillus cereus*	14	3.4
*Corynebacterium striatum*	6	1.4
*Bacillus spp.*	3	0.7
*Propionibacterium acnes*	2	0.5
*Listeria monocytogenes*	1	0.2
**Gram-negative bacilli**		
*Klebsiella pneumoniae*	14	3.4
*Enterobacter cloacae*	14	3.4
*Serratia marcescens*	12	2.9
*Pseudomonas aeruginosa*	12	2.9
*Klebsiella oxytoca*	7	1.7
*Escherichia coli*	7	1.7
*Acinetobacter baumannii complex*	6	1.4
*Escherichia coli* (ESBL)	6	1.4
*Enterobacter aerogenes*	4	1.0
*Stenotrophomonas maltophilia*	4	1.0
*Burkholderia cepacia*	3	0.7
*Enterobacter aerogenes* (CRE)	3	0.7
*Alcaligenes xylosoxidans*	1	0.2
*Campylobacter coli*	1	0.2
*Edwardsiella tarda*	1	0.2
*Eikenella corrodens*	1	0.2
*Proteus mirabilis (ESBL)*	1	0.2
*Pseudomonas putida*	1	0.2
*Ralstonia mannitolilytica*	1	0.2
*Serratia liquefaciens*	1	0.2
**Fungi**		
*Candida albicans*	30	7.2
*Candida parapsilosis*	21	5.1
*Candida guilliermondii*	5	1.2
*Candida glabrata*	5	1.2
*Candida tropicalis*	3	0.7
*Candida famata*	1	0.2
*Candida lusitaniae*	1	0.2
*Candida krusei*	1	0.2
*Cryptococcus neoformans*	1	0.2
Total	414	100

BSI, bloodstream infection; MSSA, methicillin-susceptible *Staphylococcus aureus*; MRSA, methicillin-resistant *Staphylococcus aureus*; ESBL, extended-spectrum β-lactamase; CRE, carbapenem-resistant *Enterobacterales*.

**Table 2 pathogens-12-00212-t002:** Differences in the clinical characteristics of persistent bacteremia caused by gram-positive cocci, gram-negative rods, and *Candida* species.

	GPC Group (n = 220)	GNR Group (n = 100)	*Candida* spp. Group (n = 67)	*p*-Value	Corrected *p*-Value		
					GPC Group vs. GNR Group	GPC Group vs. *Candida* spp. Group	GNR Group vs. *Candida* spp. Group
**Demography**							
Sex (male, %)	138 (62.7)	67 (67.0)	40 (59.7)				
Age, years, median (IQR)	59.0 (51.5–67.5)	67.0 (50.0–69.0)	68.0 (57.0–68.0)	0.004	0.005	0.021	
**Comorbidities**							
Diabetes mellitus	41 (18.6)	15 (15.0)	11 (16.4)				
Renal failure	112 (50.9)	58 (58.0)	30 (44.8)				
ESDR on hemodialysis	24 (10.9)	6 (6.0)	1 (1.5)	0.026		0.039	
Liver cirrhosis	23 (10.5)	5 (5.0)	4 (6.0)				
Solid malignancy	62 (28.2)	29 (29.0)	29 (43.3)				
Hematologic malignancy	20 (9.1)	6 (6.0)	4 (6.0)				
Neutropenia	9 (4.1)	4 (4.0)	5 (7.5)				
Immunosuppression	43 (19.5)	16 (16.0)	14 (20.9)				
**Severities**							
Respiratory failure	61 (27.7)	44 (44.0)	17 (25.4)	<0.001	0.014		0.030
Shock	21 (9.5)	15 (15.0)	4 (6.0)				
**Vital signs**							
BMI, kg/m^2^, median (IQR)	21.8 (18.7–24.3)	21.4 (19.0–24.8)	19.8 (17.3–23.2)				
Body temperature, °C, median (IQR)	38.1 (37.5–39.0) (n = 200)	38.7 (37.9–39.2) (n = 96)	38.6 (37.6–39.1) (n = 59)				
**Laboratory markers**							
White blood cell count, 10⁹/L, median (IQR)	9000.0 (5900.0–11,900.0)	9300.0 (5750.0–13,625.0)	7300.0 (5500.0–12,750.0)				
Neutrophil count, 10⁹/L, median (IQR)	7255.0 (4595.0–10,555.0) (n = 208)	8470.0 (4422.5–12,675.0)	6255.0 (4145.0–10,977.5) (n = 66)				
C-reactive protein, mg/dL, median (IQR)	6.8 (3.1–12.8)	8.9 (3.8–14.7)	7.7 (3.1–11.6)				
**Devices**							
Intravascular device	137 (62.3)	64 (64.0)	56 (83.6)	0.003		0.003	0.016
Intravascular device removal	111 (81.0)	52 (81.3)	47 (83.9)				
CIED	22 (10.0)	5 (5.0)	0 (0)	0.005		0.010	0.021
Ventricular assist device	9 (4.1)	8 (8.0)	1 (1.5)				
Valve replacement	26 (11.8)	7 (7.0)	2 (3.0)				
Vascular graft	16 (7.3)	20 (20.0)	6 (9.0)	0.004	0.005		
ECMO	3 (1.4)	3 (3.0)	7 (10.4)	0.031		0.006	
Continuous hemodiafiltration	37 (16.8)	24 (24.0)	14 (20.9)				
Mechanical ventilation	57 (25.9)	43 (43.0)	21 (31.3)	0.010	0.008		
**Status of persistent bacteremia**							
The period until FUBC is carried out, median (IQR)	3.0 (2.0–4.0)	3.0 (1.0–4.0)	4.0 (2.0–5.5)	0.014		0.016	0.028
Duration of bacteremia, median (IQR)	3.0 (2.0–7.0)	4.0 (2.0–7.0)	5.0 (3.0–7.5)	0.003		0.002	0.019
The ratio of persistent bacteremia	220 (11.1)	100 (12.1)	67 (29.6)	<0.001		<0.001	<0.001
**Site of infection**							
CRBSI	115 (40.9)	37 (31.9)	47 (58.8)	<0.001	0.112	0.010	0.001
Infectious endocarditis	31 (11.0)	2 (1.7)	1 (1.3)	<0.001	0.004	0.007	
Septic embolism	15 (5.3)	2 (1.7)	1 (1.3)				
Endovascular devices infections	18 (6.4)	5 (4.3)	3 (3.8)				
Thrombophlebitis	15 (5.3)	10 (8.6)	5 (6.3)				
Pyogenic spondylitis	14 (5.0)	1 (0.9)	0 (0)	0.023			
Abscess	18 (6.4)	6 (5.2)	3 (3.8)				
Pneumonia	0 (0)	2 (1.7)	0 (0)				
Intra-abdominal infections	3 (1.1)	4 (3.4)	1 (1.3)				
Urinary tract infections	2 (0.7)	10 (8.6)	2 (2.5)	<0.001	<0.001		
Biliary tract infections	1 (0.4)	2 (1.7)	0 (0)				
Skin and soft tissue infections	5 (1.8)	1 (0.9)	1 (1.3)				
Surgical site infection	6 (2.1)	7 (6.0)	0 (0)	0.028			
Others	11 (3.9)	2 (1.7)	6 (7.5)				
Primary	27 (9.6)	25 (21.6)	10 (12.5)	0.007	0.008		
**Hospital** **stays**							
Duration of hospitalization, days, median (IQR)	64.0 (38.0–114.0) (n = 219)	76.0 (37.5–150.3)	71.5 (43.3–112.0) (n = 66)				
Presence of ICU	86 (39.1)	57 (57.0)	28 (41.8)	0.011	0.011		
Duration of ICU stay, days, median (IQR)	0 (0–10.3)	4.0 (0–45.0)	0 (0–15.0)	0.001	<0.001		0.034
Presence of HCU	14 (6.4)	13 (13.0)	5 (7.5)				
Duration of HCU stay, days, median (IQR)	0 (0–0)	0 (0–0)	0 (0–0)				
Presence of CCU	9 (4.1)	3 (3.0)	1 (1.5)				
Duration of CCU stay, days, median (IQR)	0 (0–0)	0 (0–0)	0 (0–0)				
**Intervention**							
The use of antibiotics (Appropriate)	191 (86.8)	89 (89.0)	62 (92.5)				
Source control	136 (61.8)	52 (52.0)	45 (67.2)				

Data are presented as number (%) unless indicated otherwise. *p*-value: *p*-value of Kruskal-Wallis tests for continuous variables and Fisher’s exact tests for categorical variables. Corrected *p*-value: *p*-value of Steel-Dwass tests for continuous variables and Fisher’s exact test for categorical variables corrected by Holm-Bonferroni method. *p*-values were listed only for those that showed significant differences. Immunosuppression was considered in the presence of neutropenia, hematopoietic stem cell transplantation, solid organ transplantation, and corticosteroid therapy (prednisone 16 mg/day for 15 days). Intravascular devices contain a central line, such as conventional central venous catheter, peripherally inserted central catheter, tunneled central venous catheter, and implanted central venous port. BMI, body mass index; CCU, coronary care unit; CIED, cardiac implantable electronic devices; CRBSI, catheter-related blood stream infection; ECMO, extracorporeal membrane oxygenation; ESDR, end-stage renal disease; FUBC, follow-up blood culture; GPC, gram-positive cocci; GNRs, gram-negative rods; HCU, high care unit; ICU, intensive care unit; IQR, interquartile range.

**Table 3 pathogens-12-00212-t003:** Differences in clinical characteristics between patients with and without clearance of persistent bacteremia.

	GPC (n = 220)		Odds Ratio [95% CI]	*p*-Value	GNR (n = 100)		Odds Ratio [95% CI]	*p*- Value	*Candida* spp. (n = 67)		Odds Ratio [95% CI]	*p*- Value
	PB Clearance Group (n = 172)	PB without Clearance Group (n = 48)			PB Clearance Group (n = 72)	PB without Clearance Group (n = 28)			PB Clearance Group (n = 46)	PB without Clearance Group (n = 21)		
**Comorbidities**												
Diabetes mellitus	30 (17.4)	11 (22.9)	0.7 [0.3–1.6]		10 (13.9)	5 (17.9)	0.7 [0.2–2.4]		9 (19.6)	2 (9.5)	2.3 [0.5–11.8]	
Renal failure	87 (50.6)	25 (52.1)	0.9 [0.5–1.8]		41 (56.9)	17 (60.7)	0.9 [0.4–2.1]		19 (41.3)	11 (52.4)	0.6 [0.2–1.8]	
ESDR on hemodialysis	20 (11.6)	4 (8.3)	1.5 [0.5–4.5]		4 (5.6)	2 (7.1)	0.8 [0.1–4.4]		1 (2.2)	0 (0)	–	
Liver cirrhosis	20 (11.6)	3 (6.3)	2 [0.6–7]		3 (4.2)	2 (7.1)	0.6 [0.1–3.6]		4 (8.7)	0 (0)	–	
Solid malignancy	48 (27.9)	14 (29.2)	1 [0.5–1.9]		18 (25.0)	11 (39.3)	0.5 [0.2–1.3]		21 (45.7)	8 (38.1)	1.4 [0.5–3.9]	
Hematologic malignancy	15 (8.7)	5 (10.4)	0.8 [0.3–2.4]		4 (5.6)	2 (7.1)	0.8 [0.1–4.4]		2 (4.3)	2 (9.5)	0.4 [0.1–3.3]	
Neutropenia	8 (4.7)	1 (2.1)	1.1 [0.2–5.5]		2 (2.8)	2 (7.1)	0.4 [0.1–2.8]		2 (4.3)	3 (14.3)	0.3 [0–1.8]	
Immunosuppression	31 (18.0)	12 (25.0)	0.7 [0.3–1.4]		9 (12.5)	7 (25.0)	0.4 [0.1–1.3]		6 (13.0)	8 (38.1)	0.2 [0.1–0.8]	0.027
**Severities**												
Respiratory failure	51 (29.7)	10 (20.8)	1.6 [0.7–3.5]		31 (43.1)	13 (46.4)	0.9 [0.4–2.1]		10 (21.7)	7 (33.3)	0.6 [0.2–1.7]	
Shock	15 (8.7)	6 (12.5)	0.7 [0.2–1.8]		11 (15.3)	4 (14.3)	1.1 [0.3–3.7]		3 (6.5)	1 (4.8)	1.4 [0.1–14.3]	
**Devices**												
Intravascular device	102 (59.3)	35 (72.9)	0.5 [0.3–1.1]		48 (66.7)	16 (57.1)	1.5 [0.6–3.7]		38 (82.6)	18 (85.7)	0.8 [0.2–3.3]	
CIED	19 (11.0)	3 (6.3)	1.9 [0.5–6.6]		5 (6.9)	0 (0)	–		0 (0)	0 (0)	–	
Ventricular assist device	7 (4.1)	2 (4.2)	1 [0.2–4.9]		7 (9.7)	1 (3.6)	2.9 [0.3–24.8]		1 (2.2)	0 (0)	–	
Valve replacement	24 (14.0)	2 (4.2)	3.7 [0.9–16.4]		6 (8.3)	1 (3.6)	2.5 [0.3–21.4]		1 (2.2)	1 (4.8)	0.4 [0–7.5]	
Vascular graft	11 (6.4)	5 (10.4)	0.6 [0.2–1.8]		14 (19.4)	6 (21.4)	0.9 [0.3–2.6]		5 (10.9)	1 (4.8)	2.4 [0.3–22.3]	
ECMO	2 (1.2)	1 (2.1)	0.6 [0–6.2]		2 (2.8)	1 (3.6)	0.8 [0.1–8.9]		1 (2.2)	6 (28.6)	0.1 [0–0.5]	0.003
Continuous hemodiafiltration	30 (17.4)	7 (14.6)	1.2 [0.5–3]		19 (26.4)	5 (17.9)	1.7 [0.6–5]		8 (17.4)	6 (28.6)	0.5 [0.2–1.8]	
Mechanical ventilation	45 (26.2)	12 (25.0)	1.1 [0.5–2.2]		34 (47.2)	9 (32.1)	1.9 [0.8–4.7]		16 (34.8)	5 (23.8)	1.7 [0.5–5.5]	
**Site of infection**												
CRBSI	84 (38.0)	31 (51.7)	0.6 [0.3–1]		27 (31.8)	10 (32.2)	1 [0.4–2.4]		33 (55.9)	14 (66.7)	0.6 [0.2–1.8]	
Infectious endocarditis	30 (13.6)	1 (1.7)	9.3 [1.2–69.4]	0.005	2 (2.4)	0 (0)	–		1 (1.7)	0 (0)	–	
Septic embolism	11 (5.0)	4 (6.7)	0.7 [0.2–2.4]		2 (2.4)	0 (0)	–		1 (1.7)	0 (0)	–	
Endovascular devices infections	14 (6.3)	4 (6.7)	1 [0.3–3]		5 (5.9)	0 (0)	–		3 (5.1)	0 (0)	–	
Thrombophlebitis	14 (6.3)	1 (1.7)	4 [0.5–31]		9 (10.6)	1 (3.2)	3.6 [0.4–29.3]		5 (8.5)	0 (0)	–	
Pyogenic spondylitis	11 (5.0)	3 (5.0)	1 [0.3–3.7]		1 (1.2)	0 (0)	–		0 (0)	0 (0)	–	
Abscess	16 (7.2)	2 (3.3)	2.3 [0.5–10.1]		6 (7.1)	0 (0)	–		2 (3.4)	1 (4.8)	0.7 [0.1–8.2]	
Pneumonia	0 (0)	0 (0)	–		2 (2.4)	0 (0)	–		0 (0)	0 (0)	–	
Intra-abdominal infections	2 (0.9)	1 (1.7)	0.5 [0–6]		3 (3.5)	1 (3.2)	1.1 [0.1–11]		1 (1.7)	0 (0)	–	
Urinary tract infections	2 (0.9)	0 (0)	–		6 (7.1)	4 (12.9)	0.5 [0.1–2]		2 (3.4)	0 (0)	–	
Biliary tract infections	0 (0)	1 (1.7)	0		1 (1.2)	1 (3.2)	0.4 [0–5.9]		0 (0)	0 (0)	–	
Skin and soft tissue infections	3 (1.4)	2 (3.3)	0.4 [0.1–2.4]		0 (0)	1 (3.2)	0		1 (1.7)	0 (0)	–	
Surgical site infection	4 (1.8)	2 (3.3)	0.5 [0.1–3]		4 (4.7)	3 (9.7)	0.5 [0.1–2.2]		0 (0)	0 (0)	–	
Others	8 (3.6)	3 (5.0)	0.7 [0.2–2.8]		1 (1.2)	1 (3.2)	0.4 [0–5.9]		5 (8.5)	1 (4.8)	1.9 [0.2–16.8]	
Primary	22 (10.0)	5 (8.3)	1.2 [0.4–3.4]		16 (18.8)	9 (29.0)	0.6 [0.2–1.5]		5 (8.5)	5 (23.8)	0.3 [0.1–1.2]	
**Intervention**												
The use of antibiotics (Appropriate)	147 (85.5)	44 (91.7)	0.6 [0.2–1.8]		64 (88.9)	25 (89.3)	1.1 [0.3–4.7]		41 (89.1)	21 (100)	0	
Source control	102 (59.3)	34 (70.8)	0.6 [0.3–1.2]		41 (56.9)	11 (39.3)	2 [0.8–5]		33 (71.7)	12 (57.1)	1.9 [0.7–5.6]	

Data are presented as number (%) unless indicated otherwise. *p*-value: *p*-value of Kruskal-Wallis tests for continuous variables and Fisher’s exact tests for categorical variables. *p*-values were listed only for those that showed significant differences. Immunosuppression was considered in the presence of neutropenia, hematopoietic stem-cell transplantation, solid organ transplantation, and corticosteroid therapy (prednisone 16 mg/day for 15 days). Intravascular devices contain a central line, such as a conventional central venous catheter, peripherally inserted central catheter, tunneled central venous catheter, and implanted central venous port. BMI, body mass index; CCU, Coronary care unit; CIED, cardiac implantable electronic devices; CRBSI, catheter-related blood stream infection; ECMO, extracorporeal membrane oxygenation; ESDR, end-stage renal disease; FUBC, follow up blood culture; GPC, gram-positive cocci; GNRs, gram-negative rods; HCU, high care unit; ICU, intensive care unit; IQR, interquartile range.

## Data Availability

The datasets created and analyzed during the current study are not publicly available due to contain a great deal of detailed patient information. The dataset is owned by the Department of Infectious Diseases, Internal Medicine, Tohoku University Graduate School.

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
