# Peer review of "Clinical and Epidemiological Characteristics of Persistent Bacteremia: A Decadal Observational Study"

_pathogens, 2023, doi:10.3390/pathogens12020212_

Round 1

Reviewer 1 Report (Previous Reviewer 2)

I have no further comments.

Author Response

Reviewer 1

I have no further comments.

Response: We express our deep gratitude for your time and effort, despite your busy schedule, in reviewing the contents of this paper. We shall make every endeavor to enhance the content of this paper following revisions of the points raised by the other reviewers.

Reviewer 2 Report (New Reviewer)

In this manuscript, Kitaya et al. studied Clinical and epidemiological characteristics of persistent bacteremia. This retrospective observational study was performed at Tohoku University Hospital between January 2012 and December 2021.

This paper provides valuable data that the patients with bloodstream infections due to Candida spp. have a higher mortality rate than patients with persistent bacteremia (PB) due to gram-positive cocci (GPC) or gram-negative rods (GNR), and confirming the clearance of PB are useful to improve the survival rate.

1-              Figure 1 shows that among the 6,709 cases with a positive result at the initial BC, 1922 episodes were excluded because FUBC were not performed. This study has the bias that the information in the medical records didn't clearly elucidate the reasons why FUBC was performed or wasn't performed.

Author Response

Reviewer 2

  • Figure 1 shows that among the 6,709 cases with a positive result at the initial BC, 1922 episodes were excluded because FUBC were not performed. This study has the bias that the information in the medical records didn't clearly elucidate the reasons why FUBC was performed or wasn't performed.

Response: We extend our appreciation to the reviewer for their insightful commentary. As the reviewer astutely identified, a limitation of this study is the lack of information regarding the justification for or against the administration of FUBCs. Considering this, we have incorporated the following caveat into the manuscript as follows;

“Secondly, the data contained within the medical records was insufficient to fully elucidate the underlying reasons for the decision to either perform or abstain from performing FUBC.” (Lines 482–484)

Reviewer 3 Report (New Reviewer)

Bloodstream infection carries a high fatality rate. Persistent bacteremia (PB) has been shown associated with increased length of hospitalization and high mortality.

This study aimed to describe the clinical and epidemiological characteristics of PB in a tertiary University hospital from 2012 to 2021. The topic is important. However, the content should be enriched and here are comments to be addressed.

Specific comments:

1.      Figure 1. The description of “episode” in figure legend is not consistent with the definition in Methods. In addition, among the 5046 episodes, a total of 4612 episodes was excluded. It means 434 PB episodes (not 414 episodes) were included for further analysis. And, how come there were “1710 single counts”? It is suggested to check the number.

2.      Table 1. There are some organisms listed as “others”. The authors are expected to provide more information about these. For example, was there any PB episode caused by Salmonella spp.?

3.      Table 2. The age distribution between GNR group and Candida group was statistically significant (p=0.021). Please check.

4.      Table 2. Please add N (%) for “Comorbidities”, “Devices”, “Status of PB”… etc.

5.      Table 2. Why were respiratory failure and shock categorized in Comorbidities?

6.      Results. In addition to the frequency of GPC, GNR and Candida spp., the authors are expected to provide more information of organism species in the important diseases. For example, what were the common organisms caused IE (Line 255)? Are these causative organisms similar to previous reports?

7.      Results. For the vascular graft and infections after cardiovascular surgery, what were the common pathogens in this study? If the pathogens were different from previous reports, more discussion is needed.

8.      Page 9, Line 284. “GNR group (44 cases, 44.4%;…)”. It should be 44.0%.

Author Response

Reviewer 3

  1. Figure 1. The description of “episode” in figure legend is not consistent with the definition in Methods. In addition, among the 5046 episodes, a total of 4612 episodes was excluded. It means 434 PB episodes (not 414 episodes) were included for further analysis. And, how come there were “1710 single counts”? It is suggested to check the number.

Response: We express our heartfelt appreciation to the reviewer for their perceptive recommendation. As the reviewer rightly pointed out, the portrayal of the episode of bacteremia in the text was ambiguous and prone to misinterpretation, and has been rectified as follows;

“In instances where the same pathogenic agent was cultured across multiple BCs, a diagnosis of bacteremia was confirmed.” (Lines 130–131)

We have also incorporated definitions of the terms 'episode' and 'single count' within the body of the text, as well as definitions of FUBC within the captions of Figure 1, to facilitate comprehension for the reader.

“For PB, multiple consecutive BCs that identified the same strain were counted as one episode. The single counts represent the number of BCs itself, and the episodes represent a series of BSI episodes.” (Lines 136–138)

“If a microorganism of the same species as the initial one was detected in at least one pair of FUBCs, the case was classified as FUBC positive. Conversely, if disparate species were identified in the FUBC, the episode of bacteremia was deemed to be a novel occurrence.” (Annotations of figure 1)

Initially, 6,709 individual cases were found to be BC-positive, comprising a total of 5,046 episodes and 1,663 single counts (including second and subsequent positive BCs in cases of PB and second and subsequent positive BCs in cases in which FUBC detected a different strain than the first BC). It is worth noting that the annotation of 1,710 single counts in Figure 1 is inaccurate; the correct value is 1,663 single counts, which has been rectified accordingly. We are appreciative of your attention to this matter.

  1. Table 1. There are some organisms listed as “others”. The authors are expected to provide more information about these. For example, was there any PB episode caused by Salmonella spp.?

Response: We extend our sincere appreciation to the reviewer for their valuable feedback. We recognize the importance of providing a comprehensive list of all bacteria species detected, and have therefore included them in Table 1.

  1. Table 2. The age distribution between GNR group and Candida group was statistically significant (p=0.021). Please check.

Response: We extend our appreciation for your insightful recommendations pertaining to our manuscript. I checked and p=0.021 is the p value for the GPC group vs. Candida spp. group, and I made a mistake in the column where it is listed. The description has been corrected.

  1. Table 2. Please add N (%) for “Comorbidities”, “Devices”, “Status of PB”… etc.

Response: We express our appreciation for bringing this to our attention. As a result of the numerous instances in Tables 2 and 3 where the inclusion of N (%) was deemed appropriate, we have included the following notation in these tables:" Data are presented as number (%) unless indicated otherwise."

  1. Table 2. Why were respiratory failure and shock categorized in Comorbidities?

Response: We extend our gratitude for your observations regarding our manuscript. We concur with your assertion that the inclusion of respiratory failure and shock as comorbidities is incongruous, and in light of previous literature, we have chosen to classify them as severities.

  1. Results. In addition to the frequency of GPC, GNR and Candida spp., the authors are expected to provide more information of organism species in the important diseases. For example, what were the common organisms caused IE (Line 255)? Are these causative organisms similar to previous reports?

Response: We express our gratitude for your discerning recommendations pertaining to our manuscript. We recognize the clinical significance of determining whether the causative organisms of IE detected in this study are consistent with those reported in previous studies. Therefore, we have included a listing of the causative organisms of IE in this study within the text and have conducted a comparison with previous studies;

“In this investigation, methicillin-sensitive S. aureus (MSSA) (nine cases, 26.4%), Enterococcus faecalis (six cases, 17.6%), and methicillin-resistant S. aureus (MRSA) (five cases, 14.7%) were the most prevalent causative agents of IE, in concordance with previously published research.” (Lines 260–263)

  1. Results. For the vascular graft and infections after cardiovascular surgery, what were the common pathogens in this study? If the pathogens were different from previous reports, more discussion is needed.

Response: We extend our appreciation for your suggestion. Indeed, infections involving vascular grafts and those following cardiovascular surgery are frequently severe, and the identification of causative organisms is of paramount clinical importance. As patients with vascular grafts in this study did not necessarily exhibit bacteremia resulting from infections involving the grafts, we have chosen to include a listing of the causative organisms of infections following cardiovascular surgery in the text to facilitate comprehension for the reader and have compared them with previously reported studies.

“On the other hand, after cardiovascular surgery, in general the most common cause of SSIs (ie, in approximately 80% of all cases) are GPC, especially S. aureus [44]. In our hospital, GNR (Enterobacter cloacae, Klebsiella oxytoca, and Burkholderia cepacia) were the causative pathogens of SSI in a large proportion of patients after cardiovascular surgery (three cases, 60.0%). Our hospital performs a large number of cardiovascular operations, and the fact that GNR is a frequent cause of PB caused by SSI after cardiovascular surgery is considered to be one of the reasons why GNR was a frequent cause of PB caused by SSI in our hospital.” (Lines 327–331)

“At our hospital, about 70% of patients admitted to the ICU are postoperative patients after cardiovascular surgery. Such patients are often intubated and artificially ventilated, which involves frequent aspiration procedures for the prevention and treatment of VAP [51]. Dispersal of sputum containing GNR through the aspiration procedure may have contributed to the higher rate of healthcare-associated infections caused by GNR in the ICU. Additionally, it is feasible that GNR-laden sputum dispersed by suction and adhering to surgical incisions may have played a role in the high incidence of GNR as a causative agent of postoperative cardiovascular infections in our hospital.” (Lines 348–355)

  1. Page 9, Line 284. “GNR group (44 cases, 44.4%;…)”. It should be 44.0%.

Response: We extend our appreciation for your suggestion. The description was in error and has been rectified as per your indication.

This manuscript is a resubmission of an earlier submission. The following is a list of the peer review reports and author responses from that submission.

Round 1

Reviewer 1 Report

The purpose of this 10-year study was to describe the characteristics and prognosis of persistent bacteremia caused by three different pathogens.There are many important issues that could be improved.1. There is a lot of confusion about patient selection (Figure 1). eg. single count, episode, duplicate episode.  (Table1-3) Neutropenia, Immunosuppression, many devices, ? abscess, high care unit, sites of infection (Primary ?) Infectious endocarditis or infective endocarditis 2. There are many important critical conditions that are not described, including respiratory failure, intubation, acute renal failure, and shock. It is a worthwhile article, but the data could be better organized. It is strongly recommended that the authors should rewrite this article. I will evaluate this revised version again. 

Author Response

Major

  1. There is a lot of confusion about patient selection (Figure 1). eg. single count, episode, duplicate episode. (Table1-3) Neutropenia, Immunosuppression, many devices, ? abscess, high care unit, sites of infection (Primary ?) Infectious endocarditis or infective endocarditis

Response: We strongly appreciate the reviewer's comments. The single counts represent the number of blood cultures itself, and the episodes represent a series of bloodstream infection episodes. This description has been added to figure legend of figure 1. Duplicate episode indicates a duplicate case (e.g., a case that is under 18 years of age and the result is contamination).

Neutropenia, immunosuppression, and device insertion have been reported to occur more frequently in patients with bacteremia, so we included these items to determine whether they are also present in patients with persistent bacteremia. Definitions of neutropenia, immunosuppression are given on page 3 and line 142-144 and page 3 and line 144-146, respectively. The admission to a high care unit was included to confirm the difference between high care unit and intensive care unit, since admission to an intensive care unit is said to be a risk for bacteremia and increased mortality, according to a previous report. Abscess, infectious endocarditis is also included because it is considered important as a focus of infection in bacteremia. Primary indicates the cases in which the infectious focus could not be identified, and definitions are described on page 3 and line 125-127 in the text.

  1. There are many important critical conditions that are not described, including respiratory failure, intubation, acute renal failure, and shock. It is a worthwhile article, but the data could be better organized. It is strongly recommended that the authors should rewrite this article. I will evaluate this revised version again.

Response: Thank you for your valuable suggestions on our manuscript. With regard to clinical information, the decision was based on previous papers on persistent bacteremia (e.g.,Chong, Y.P. et al. Persistent Staphylococcus aureus bacteremia: a prospective analysis of risk factors, outcomes, and microbiologic and genotypic characteristics of isolates. Medicine (Baltimore) 2013, 92, 98-108, Maskarinec, S.A. et al. Positive follow-up blood cultures identify high mortality risk among patients with Gram-negative bacteraemia. Clin Microbiol Infect 2020, 26, 904-910.). In this study, we obtained the following data from the medical records and the database of the Department of Infectious Diseases: sex, age, comorbidities, body mass index, and body temperature; blood test results, including levels of serum white blood cells, neutrophils, and C-reactive protein; presence of intravascular devices, ie, a central line such as conventional central venous catheter (CVC), peripheral inserted central catheter, tunneled central venous catheter, and implanted central venous port (CV port), and removal of such devices; presence of cardiac implantable electronic devices (CIEDs), such as cardiac pacemakers, implantable cardioverter defibrillators, and devices for cardiac resynchronization therapy; presence of ventricular assist device; history of valve replacement; presence of vascular graft; use of extracorporeal membrane oxygenation (ECMO), continuous hemodiafiltration, and mechanical ventilation; interval between initial BC and FUBC; duration of bacteremia; site of infections; duration of hospitalization; whether time was spent in an ICU, high care unit, or coronary care unit between the initial BC and the last FUBC; antibiotic use; performance of source control; and mortality, recorded as early (30-day), late (30-90 day), and 90-day mortality.

The points raised by reviewer 1 are important but were not part of the analysis of our study, and these data were not available from the database we used. However, we have examined mechanical ventilation and ESDR on hemodialysis, and we believe we were able to evaluate key clinical features related to PB in this study.

Reviewer 2 Report

Comments for the authors

In the manuscript Clinical and epidemiological characteristics of persistent bacteremia: a decadal observational study”, the authors investigated the differences in clinical characteristics of persistent bacteremia between three groups of bacterial species: gram-positive cocci, gram-negative rods, and Candida spp. and the differences in mortality in presence or absence of clearance of persistent bacteremia.

Overall, the results of the study are quite interesting and the analysis carried out seems very accurate.

Here are some suggestions for perfecting the manuscript

Major revisions

Methods

Study design and setting

1.                  Page 3, line 104. ″…recorded as early (30-day), late (30-90 day), and 90-day mortality.″. In cases of late (30-90 day) and 90-day mortality, 90-day mortality cases would be counted twice by placing them in both groups. Did the authors want to indicate the third group as "over 90 days"?

In this respect, the entire text should be corrected.

Results and Discussion

Patient selection and causative pathogens of PB

2.                  I would recommend the authors to separate the “Results” section from “Discussion”. This makes reading and understanding for the reader easier.

3.                  Page 4, line 88. The causative pathogens of PB are shown in Table 2.″. Tables and figures must be cited in order of appearance. Rearrange with respect to what is reported in the text.

4.                  Pag. 5, line 191. ″BC, blood culture; FUBC, follow-up blood culture. 1,710 single counts, including second and subsequent positive blood cultures in cases of persistent bacteremia and second and subsequent positive blood cultures in cases in which follow-up blood culture detected a different strain than the first blood culture″. Include it in the comment on the figure.

5.                  Table 1. Please, adjust the formatting of the first column of the Table 1.

6.                  Table 2. Please, adjust the formatting of the Table 2.

7.                  Page 8, line 260. “1). Clinical characteristics of PB caused by GNR”. Please, correct the paragraph number.

8.                  Figures 2 and 3. Please, adjust the comments of figures.

9.                  “Competing interests: The authors declare no relevant conflicts of interest.” The authors need to clarify this aspect. We cannot speak of "non relevant" conflicts.

Please, modify according to journal instructions: “Conflicts of Interest: Declare conflicts of interest or state “The authors declare no conflict of interest.” Authors must identify and declare any personal circumstances or interest that may be perceived as inappropriately influencing the representation or interpretation of reported research results. Any role of the funders in the design of the study; in the collection, analyses or interpretation of data; in the writing of the manuscript; or in the decision to publish the results must be declared in this section. If there is no role, please state “The funders had no role in the design of the study; in the collection, analyses, or interpretation of data; in the writing of the manuscript; or in the decision to publish the results”.”

10.              “Availability of data and materials: The datasets used and/or analyzed in this study are available 490 from the corresponding author upon reasonable request.” The authors should declare with which Institution the data are preserved also for reasons of privacy.

Author Response

Major

Methods

Study design and setting

  1. Page 3, line 104. ″…recorded as early (30-day), late (30-90 day), and 90-day mortality.″. In cases of late (30-90 day) and 90-day mortality, 90-day mortality cases would be counted twice by placing them in both groups. Did the authors want to indicate the third group as "over 90 days"?

In this respect, the entire text should be corrected.

Response: We would like to thank the reviewer for the insightful comment. When comparing mortality rates among the three strains, we decided that it would be easier for readers to understand more clearly if mortality rates for the entire 90-day period were also included, so we have specified 90-day mortality rates in addition to early and late mortality rates.

Results and Discussion

Patient selection and causative pathogens of PB

  1. I would recommend the authors to separate the “Results” section from “Discussion”. This makes reading and understanding for the reader easier.

Response: Thank you for your recommendation. Since many items were examined in this study, it was considered that it might be difficult for readers to understand if each item was described separately in the results and discussion sections. Therefore, to make it easier for the reader to read, the results and discussion are combined and organized by each theme.

  1. Page 4, line 88. ″ The causative pathogens of PB are shown in Table 2.″. Tables and figures must be cited in order of appearance. Rearrange with respect to what is reported in the text.

Response: We would like to thank the reviewer for the comment. As you indicated, we have rearranged the order of the tables.

  1. Pag. 5, line 191. ″BC, blood culture; FUBC, follow-up blood culture. ※1,710 single counts, including second and subsequent positive blood cultures in cases of persistent bacteremia and second and subsequent positive blood cultures in cases in which follow-up blood culture detected a different strain than the first blood culture″. Include it in the comment on the figure.

Response: We would like to thank the reviewer for the constructive comment. We have included the text you pointed out in the comments on the figure.

  1. Table 1. Please, adjust the formatting of the first column of the Table 1.

Response: We would like to thank the reviewer for the comment. We have corrected formatting of the first column of the Table 1.

  1. Table 2. Please, adjust the formatting of the Table 2.

Response: Thank you once again. We have corrected formatting of the first column of the Table 2.

  1. Page 8, line 260. “1). Clinical characteristics of PB caused by GNR”. Please, correct the paragraph number.

Response: We would like to thank the reviewer for the insightful comment. We have corrected the paragraph numbers in the section you indicated.

  1. Figures 2 and 3. Please, adjust the comments of figures.

Response: We would like to thank the reviewer for the comment. We have adjusted the comments of figures.

  1. “Competing interests: The authors declare no relevant conflicts of interest.” The authors need to clarify this aspect. We cannot speak of "non relevant" conflicts. Please, modify according to journal instructions: “Conflicts of Interest: Declare conflicts of interest or state “The authors declare no conflict of interest.” Authors must identify and declare any personal circumstances or interest that may be perceived as inappropriately influencing the representation or interpretation of reported research results. Any role of the funders in the design of the study; in the collection, analyses or interpretation of data; in the writing of the manuscript; or in the decision to publish the results must be declared in this section. If there is no role, please state “The funders had no role in the design of the study; in the collection, analyses, or interpretation of data; in the writing of the manuscript; or in the decision to publish the results”.”

Response: We strongly appreciate the reviewer's comments. We believe that an accurate description of conflicts of interest is very important to maintain the fairness of the research. The description of Conflicts of Interest has been corrected as you indicated.

  1. “Availability of data and materials: The datasets used and/or analyzed in this study are available 490 from the corresponding author upon reasonable request.” The authors should declare with which Institution the data are preserved also for reasons of privacy.

Response: We would like to thank the reviewer for the comments. Because the results of this study, although anonymous, contain a great deal of detailed patient information, we have decided to keep the results private for privacy and ethical reasons. Therefore, we have marked it as “Not applicable”.

Round 2

Reviewer 1 Report

We appreciate the author's prompt response to our questions. My question may be unclear to the author. As a result, my question remains unanswered. A article should be well-defined, as most readers will agree. There is a difference between infectious disease and infectious disease, in my opinion. ESDR is not the same as ESRD (end-stage renal disease). I don't know if the authors can agree on whether patients with ESRD should receive hemodialysis, peritoneal dialysis or no dialysis at all. This article still needs to be rewritten, in my opinion.

Reviewer 2 Report

In my opinion, the authors, for transparency, must indicate where the data is kept. This does not mean having access to the data but giving the possibility, even in the event of subsequent checks, for any reason, on what is reported in the manuscript, to have a reference institution for any access requests.